# Metabolic Effects of New Glucose Transporter (GLUT-1) and Lactate Dehydrogenase-A (LDH-A) Inhibitors against Chemoresistant Malignant Mesothelioma

**DOI:** 10.3390/ijms24097771

**Published:** 2023-04-24

**Authors:** Marika A. Franczak, Oliwia Krol, Gabriela Harasim, Agata Jedrzejewska, Nadia Zaffaroni, Carlotta Granchi, Filippo Minutolo, Amir Avan, Elisa Giovannetti, Ryszard T. Smolenski, Godefridus J. Peters

**Affiliations:** 1Department of Biochemistry, Medical University of Gdansk, 80-210 Gdańsk, Poland; 2Department of Medical Oncology, Amsterdam University Medical Centers, Location VUmc, Cancer Center Amsterdam, 1081 HV Amsterdam, The Netherlands; 3Molecular Pharmacology Unit, Fondazione IRCCS Istituto Nazionale dei Tumori, 20133 Milano, Italy; 4Dipartimento di Farmacia, Università di Pisa, 56126 Pisa, Italy; 5Metabolic Syndrome Research Center, Mashhad University of Medical Sciences, Mashhad 91886-17871, Iran; 6Medical Genetics Research Center, Mashhad University of Medical Sciences, Mashhad 91886-17871, Iran; 7Fondazione Pisana per la Scienza, 56124 Pisa, Italy

**Keywords:** malignant mesothelioma, lactate dehydrogenase, glucose transporter type 1, chemoresistance, anticancer treatment, cancer metabolism

## Abstract

Malignant mesothelioma (MM) is a highly aggressive and resistant tumor. The prognostic role of key effectors of glycolytic metabolism in MM prompted our studies on the cytotoxicity of new inhibitors of glucose transporter type 1 (GLUT-1) and lactate dehydrogenase-A (LDH-A) in relation to ATP/NAD^+^ metabolism, glycolysis and mitochondrial respiration. The antiproliferative activity of GLUT-1 (PGL13, PGL14) and LDH-A (NHI-1, NHI-2) inhibitors, alone and in combination, were tested with the sulforhodamine-B assay in peritoneal (MESO-II, STO) and pleural (NCI-H2052 and NCI-H28) MM and non-cancerous (HMEC-1) cells. Effects on energy metabolism were measured by both analysis of nucleotides using RP-HPLC and evaluation of glycolysis and respiration parameters using a Seahorse Analyzer system. All compounds reduced the growth of MM cells in the µmolar range. Interestingly, in H2052 cells, PGL14 decreased ATP concentration from 37 to 23 and NAD^+^ from 6.5 to 2.3 nmol/mg protein. NHI-2 reduced the ATP/ADP ratio by 76%. The metabolic effects of the inhibitors were stronger in pleural MM and in combination, while in HMEC-1 ATP reduction was 10% lower compared to that of the H2052 cells, and we observed a minor influence on mitochondrial respiration. To conclude, both inhibitors showed cytotoxicity in MM cells, associated with a decrease in ATP and NAD^+^, and were synergistic in the cells with the highest metabolic modulation. This underlines cellular energy metabolism as a potential target for combined treatments in selected cases of MM.

## 1. Introduction

The metabolic processes in cancer cells are markedly distinct from those of healthy cells and this difference plays a crucial role in their survival in the host organism [1]. Indeed, this cancer-specific metabolism enables tumor cells to escape from the immune system, stimulate metastasis and become resistant to current anti-cancer therapies [2]. One of the most commonly observed differences between cancer and normal cells is the Warburg effect or aerobic glycolysis. In this metabolic switch, cells switch energy metabolism from mitochondrial oxidative phosphorylation (OXPHOS) to energy production from glucose by glycolysis, even in the presence of oxygen [3]. The Warburg effect provides neoplastic cells with increased and rapid biosynthesis of adenosine triphosphate (ATP), facilitating their proliferation and reducing risk of cell damage by reactive oxygen species (ROS). Additionally, it contributes to the creation of a highly acidic tumor microenvironment, making cancer cells more invasive [4]. Of note, rapidly proliferating non-cancer cells, such as lymphocyte T or endothelial cells, may also rely on anaerobic glycolysis as their main energy source, although this metabolic pathway is much less efficient than OXPHOS in terms of glucose consumption [5].

One of the key enzymes in glycolysis is lactate dehydrogenase (LDH) which catalyzes the transformation of pyruvate into lactate by using the reduced form of nicotinamide adenine dinucleotide (NADH) as the cofactor [1,2]. So far, six isoforms of LDH have been reported, consisting of various combinations of three different subunits (LDH-A, LDH-B and LDH-C), with LDH-A being the most commonly found subunit in various malignancies such as colon, ovarian, stomach, kidney and lung cancer [3]. The measurement of LDH levels in serum is commonly used as a predictive factor for treatment efficacy, cancer recurrence and patient survival in several cancers including melanoma, lymphoma or lung cancer [4,5,6]. Most importantly, in various cancers, including pancreatic cancer, non-small-cell lung cancer, breast cancer and mesothelioma, high expression of LDH-A is correlated with significantly shorter patient survival [7,8,9]. On the other hand, in non-cancer patients with genetic LDH-A deficiency, myoglobinuria is the only noted symptom, occurring after intense exercise [10]. Thus, given the presumed limited side effects, LDH-A inhibition represents an interesting target for anticancer treatment. Due to their high proliferation [11], cancer cells require an increased amount of glucose to meet their energy demands, which is associated with an increase in the number of glucose transporters (GLUTs) [12,13]. GLUTs are a group of transport proteins that facilitate the transportation of glucose across cell membranes. One subtype of GLUTs, GLUT-1, is predominantly found in erythrocytes, endothelial cells and cancer cells [14]. Of note, a recent meta-analysis supported the diagnostic value of GLUT-1 expression in distinguishing mesothelioma from reactive mesothelial cells [15]. Furthermore, similar to LDH-A expression, high expression of GLUT-1 has been observed in many tumors, including pancreatic, breast and lung cancers, and has been associated with chemoresistance, metastasis and significantly shorter patient survival [9,16,17].

Malignant mesothelioma (MM) is a rare and highly aggressive tumor that originates from the pleural or peritoneal cavities, causing a significant number of deaths each year [18]. In a previous study, we observed that hypoxia leads to high LDH-A expression and resistance to gemcitabine in MM cells, which was reduced by combined treatment with an LDH-A inhibitor (NHI-1) [19]. Based on these findings and the high expression of GLUT-1 in tumors, we designed and synthesized new specific inhibitors of LDH-A [20] and GLUT-1 [21], aiming to specifically target the energy metabolism in cancer cells. The two LDH-A inhibitors used in this study (NHI-1 and NHI-2, Figure 1a) belong to the chemical class of N-hydroxyindole-2-carboxylates, whereas the two GLUT-1 inhibitors (PGL13 and PGL14, Figure 1b) are salicylketoxime derivatives. The present study aimed to further characterize the effects of these inhibitors by focusing on their cytotoxicity effects in relation to purine nucleotide levels, glycolysis parameters and mitochondrial respiration in a panel of MM cells representative of the different anatomical localizations and tissues types of these tumors.

## 2. Results

### 2.1. Effect of GLUT-1 and LDH-A Inhibitors on the Growth of Mesothelioma and HMEC-1 Cells

The 50% cell growth inhibitory effect (IC_50_) of the GLUT-1 and LDH-A inhibitors was assessed using the SRB assay. Cells were incubated with GLUT-1 (PGL13, PGL14) and LDH-A (NHI-1, NHI-2) inhibitors for 72 h in normoxic conditions. The LDH-A inhibitors showed IC_50_ values of around 20–25 µM across the mesothelioma cell lines, with the exception of H2052 and MESO-II cells, which displayed the highest (43.5 ± 4.8 μM) and lowest (18.5 ± 2.0 μM) IC_50_ values for NHI-2, respectively (Figure 2a; Appendix A). These values were significantly higher compared to results obtained in hypoxic conditions (Figure 2c), which is associated with increased expression of LDH-A, as described previously [13].

The GLUT-1 inhibitor PGL14 exhibited IC_50_ values of 6.4 ± 0.2 μM in MESO-II and 44.4 ± 0.2 μM in H2052 (Figure 2a), whereas PGL13 IC_50_ values ranged from 13 to 39 μM (Appendix A).

Therefore, we selected PGL14 and NHI-2 for the following studies on the combinations of the LDH-A and GLUT-1 inhibitors in the most resistant (H2052) and sensitive (MESO-II) cells. Our previous study showed a synergistic interaction of LDH-A inhibitors with gemcitabine against MESO-II and STO cells, cultured both under normoxic and hypoxic conditions (1% O_2_). Therefore, in order to evaluate the activity on the potential aerobic conditions of resistant tumor areas, we decided to focus our combination experiments on cells growing under normoxic conditions. The cells were then treated with NHI-2 (0.1–60 μM) and PGL14 (IC_50_) or with PGL14 (0.1–60 μM) and NHI-2 (IC_50_), and the combination index values (CIs) indicated strong synergistic effects in H2052 cells and additive/synergistic effects in MESO-II cells (Figure 2d).

### 2.2. Effect of GLUT-1 and LDH-A Inhibitors, as Single Agents or in Combination, on Intracellular Nucleotide Concentrations in Mesothelioma Cell Lines

To determine the effect of the GLUT-1 and LDH-A inhibitors on the intracellular nucleotide concentrations, cells were treated with PGL13, PGL14, NHI-1 and NHI-2 at their IC_50_ concentration for 24 h in normoxia. GLUT-1 inhibitors decreased ATP and NAD^+^ levels in all cell lines except MESO-II. Of note, PGL14 was most effective in H2052 cells. Similarly, NHI-2 decreased the nucleotide concentration in H28, H2052 and MESO-II cells, while NHI-1 significantly reduced nucleotide pools only in H2052 cells (Appendix A).

To characterize the changes in cellular energy charge (ATP/ADP) and redox (NADH/NAD^+^) status, the ratios of nucleotides were calculated. The ATP/ADP ratio decreased in most cell lines, with the most pronounced effect in H2052 cells for all inhibitors. Moreover, the NADH/NAD^+^ ratio increased compared to the control in H28, H2052 and STO, but not in MESO-II cells (Figure 3).

The effect of inhibitors on energy status in non-cancer cells was tested by analyzing nucleotide levels in HMEC-1 cells after exposure to GLUT-1 and LDH-A inhibitors at concentrations close to their IC_50_ values measured in mesothelioma cells. Nucleotide pools were only slightly decreased in HMEC-1 by both the GLUT-1 inhibitors (Appendix A), whereas among the LDH-A inhibitors only NHI-2 reduced those pools (Appendix A). However, we did not observe significant modulation of the ATP/ADP ratio and NADH/NAD^+^ ratio except when exposing the HMEC-1 cells to 45 μM NHI-2 (Appendix A).

To determine whether the combination of two drugs would lead to a more pronounced effect, we treated MESO-II cells with inhibitors of both GLUT-1 and LDH-A at the IC_50_s of each inhibitor, in four combinations. The MESO-II cell line was chosen since the separate effects of each drug were relatively more modest in these cells (Appendix A). Interestingly, the combination decreased the contents of all nucleotides. The combination of PGL14 and NHI-2 produced the most pronounced effect (Figure 4a,b), suggesting that the additive/synergistic interaction of these inhibitors is associated with a reduction in both intracellular purine nucleotide and nicotinamide adenine dinucleotide pools.

### 2.3. The Effect of GLUT-1 and LDH-A Inhibitors on Glycolytic Parameters and Mitochondrial Respiration of Mesothelioma Cell Lines

In order to determine whether the effects on nucleotide concentration were related to their synthesis, we evaluated their effect on the glycolytic pathway in H2052 and MESO-II cells and on mitochondrial respiration in H2052 cells. These cells were exposed to NHI-2 and PGL14 at their IC_50_ alone and in combination for 24 h in normoxia, and then the extracellular acidification rate (ECAR, Figure 5) and oxygen consumption rate (OCR, Figure 6) were measured with an Agilent Seahorse Analyzer.

In H2052 cells, NHI-2 had a pronounced effect, whereas PGL14 only slightly increased ECAR compared to the control (Figure 5a,c). After adding the glucose, the calculated glycolysis decreased slightly following inhibition with PGL14 and combination treatments, confirming the blockage of glucose transport into cells. In addition, the other calculated glycolysis parameters, namely, glycolytic capacity and reserve, both increased upon treatment with NHI-2 and PGL14, suggesting that mitochondrial-independent glycolysis and maximum glycolytic efficiency were stimulated. However, the combination treatment lowered those parameters, implying a more pronounced effect on glycolytic processes in cells compared to single treatments. In MESO-II cells, we did not observe significant modulations, but the pattern of changes was similar to that observed in H2052 (Figure 5b,d).

To determine whether the inhibitors can also lead to a change in mitochondrial respiration, we measured the OCR in H2052 cells while considering the ECAR changes that occurred after treatment in this line. We observed only a slight reduction in mitochondrial function after a 24 h exposure to the combination in normoxia in H2052 cells (Figure 6a). In addition, a combination treatment led to a significant reduction in ATP production, which is in line with the above measurement of intracellular nucleotide concentration, and a decrease in spare respiratory capacity: a determinant of cells’ ability to react to high energy demand. In addition, the increase in proton leak after exposure to NHI-2 and the combination can be a sign of mitochondrial damage (Figure 6b,c).

## 3. Discussion

In the present study, we demonstrate that GLUT-1 and LDH-A inhibitors can hinder the growth of mesothelioma cells, and this effect can be associated with their influence on cellular metabolism (Figure 7). The cytotoxicity was observed in the µmolar range, which also led to severe disruption of the energy balance.

In previous studies, we observed that LDH-A inhibition in pleural MM cells also enhanced sensitivity to gemcitabine [19] and pemetrexed [8], supporting the hypothesis that modulation of cellular metabolism by LDH-A inhibition can increase the effectiveness of standard therapies. Nowadays, several inhibitors of LDH-A are known, such as oxamate, FX-11 and GNE140, but their preclinical results were disappointing and none of them have been clinically tested [1,22,23]. Given the documented effectiveness of GLUT-1 inhibitors in various cancer cells (lymphoma, prostate cancer, gastric cancer) [24] we examined the effect of our GLUT-1 inhibitors on mesothelioma cells. Taking the above into account, the combination of LDH-A and GLUT-1 inhibitors can enhance their influence on cancer metabolism, not only through the blockade of cellular energy production but also by limiting cellular access to the main substrate for its production.

The inhibitors of LDH-A and GLUT-1 affect the nucleotide concentrations in MM cells, especially in pleural mesothelioma. The most notable effect on cell energy was observed in the reduction of the ATP/ADP ratio and in the increase of the NADH/NAD^+^ ratio (redox status). Cells that utilize aerobic glycolysis have high ratios of ATP/ADP and NADH/NAD^+^, and any disturbances (usually reduction) in these ratios will interfere with cell growth and even lead to apoptosis [25]. Hence, our results show that both inhibitors could induce cancer cell death by affecting the intracellular nucleotide pools (Figure 7). Moreover, in peritoneal mesothelioma cells, the combined treatment with GLUT-1 and LDH-A inhibitors significantly reduced the nucleotide pool, giving new options for further study and anticancer treatments. Notably, a synergistic effect that inhibits growth and induces apoptosis was observed when imatinib was combined with a GLUT-1 inhibitor (WZB117) in imatinib-resistant gastrointestinal stromal tumor (GIST) cells, indicating the potential of such combinations for anticancer therapies [26].

In addition, it was observed that peritoneal mesothelioma cells exhibited higher glycolysis parameters compared to pleural mesothelioma cells. The insignificant changes in ECAR and OCR are in line with the observation of Lennon et al., where mitochondrial morphology changes were more visible than OCR and ECAR (measured by Seahorse) after treatment with mitochondrial inhibitors (metformin, mdivi-1) [27]. On the other hand, a recently reported LDH-A inhibitor, ML-05, reduces the ECAR and the relative ATP levels in melanoma cells (B16F10) after treatment [28], indicating its potential for further investigation with LDH-A inhibitors in other types of cancer cells. On the other hand, Jiang et al. observed an increase of ECAR and OCR in breast cancer cells (MDA-MB-231) after exposure to the LDH-A inhibitor FX11 [29]. This is similar to our results, particularly in H2052 cells.

A reference LDH-A inhibitor (oxamate) was observed to have lower cytotoxicity in normal lung epithelial cells compared to NSCLC cells by Yang et al. [30]. Apparently, the effect of GLUT-1 inhibition is less relevant as a metabolic effect (leading to cell death) than the inhibition of LDH-A. However, both inhibitors showed a more pronounced effect on nucleotide concentrations in mesothelioma cells than in non-cancer cells, suggesting that their efficacy is cancer-specific.

Interestingly, Hahne et al. observed that the loss of hypoxia-inducible factor (HIF-1) in HMEC-1 cells decreased GLUT-1, LDH-A and ATP/ADP ratio, which disrupted cancer metastasis [31]. In addition, another endothelial cell line (human umbilical vein endothelial cells; HUVEC) stopped migration and the production of tubes after being treated with resveratrol, a compound that decreased lactate production and glucose uptake [32]. These observations support our hypothesis that inhibition of GLUT-1 and LDH-A could not only affect the growth of cancer cells but also potentially inhibit metastasis.

In conclusion, LDH-A and GLUT-1 are promising targets in anticancer treatment and our results support their potential to improve cancer treatment.

## 4. Materials and Methods

### 4.1. Materials

#### 4.1.1. LDH-A and GLUT-1 Inhibitors

LDH-A inhibitors (NHI-1, NHI-2) and GLUT-1 inhibitors (PGL13, PGL14) were synthesized in the Department of Pharmacy, University of Pisa, as previously described [20,33]. The compounds were dissolved in dimethylsulfoxide (DMSO) to a 10 mM concentration. The final DMSO concentration after diluting these inhibitors for the treatment of the cells was always below 0.5%.

#### 4.1.2. Cell Culture

The human primary peritoneal mesothelioma cells (MESO-II and STO) were derived from samples of patients who underwent surgery and provided by Dr. Zaffaroni (Molecular Pharmacology Unit, Fondazione IRCCS Istituto Nazionale dei Tumori, Milano, Italy), as described previously [8]. The use of the patient samples to generate cell lines was approved by the Institutional Review Board of the Fondazione IRCCS Istituto Nazionale dei Tumori (INT). Written informed consent was obtained from all patients to donate the leftover tissue to INT after diagnostic and clinical procedures. The pleural MM (NCI-H2052, NCI-H28) cell lines and human dermal microvascular endothelial cells (HMEC-1; CRL-3243) were obtained from ATCC (Manassas, VA, USA). MESO-II and STO were cultured in Dulbecco’s Modified Eagle’s Medium F12 with 1 g/L glucose (Gibco, Thermo Fisher Scientific, Waltham, MA, USA). H28 and H2052 were cultured in RPMI 1640 medium with L-glutamine (Merck, Germany) supplemented with 10% Fetal Bovine Serum (FBS; Gibco, Thermo Fisher Scientific, Waltham, MA, USA) and 1% penicillin/streptomycin (Merck, Germany), at 37 °C, 5% CO_2_. HMEC-1 cells were cultured as described earlier [34].

### 4.2. Methods

#### 4.2.1. Analysis of Inhibition of Cell Growth

To determine the IC_50_ (concentration of inhibitor that results in 50% cell growth inhibition), 5000 mesothelioma cells were seeded per well in a 96-well plate (in 100 µL of culture medium, at 37 °C, 5% CO_2_). After 24 h (allowing cell attachment), cells were exposed to GLUT-1 inhibitors PGL13 and PGL14, and LDH-A inhibitors NHI-1 and NHI-2 by addition of 100 µL of their solutions (final concentration within the range of 0.01–60 µM). The IC_50_ was established by using the sulforhodamine-B (SRB) assay after a 72 h exposure, using the previously described method [35]. The optical densities (ODs) of exposed cells were corrected for the OD of control wells measured at day 0 (wells containing cells growing for only 24 h) and normalized to the control cells (wells with untreated cells) in order to obtain the rate of viable cells.

The effects of hypoxia on the inhibition of cell growth of MM cells treated with the LDH-A and GLUT-1 inhibitors were evaluated by growing cells at an O_2_ tension of 1%, 5% (*v/v*) CO_2_ and 94% (*v/v*) N_2_ at 37 °C, using a specific IncuSafe Jacomex Glove Box (Labo Equipment Sanyo, Loughborough, UK), as described previously [13].

The analysis of the pharmacological interaction was performed using a variable ratio (with constant IC50 concentration of the LDH-A inhibitor, NHI-2, or of the GLUT-1 inhibitor, PGL14). The cytotoxicity of these combinations was compared with the cytotoxicity of each drug alone using the combination index (CI), where CI < 0.9, CI = 0.9–1.1, and CI > 1.1 indicated synergistic, additive, and antagonistic effects, respectively. Data analysis was carried out using CalcuSyn software (Biosoft, Oxford, UK). Since we considered growth inhibition lower than 50% as not relevant, CI values at fractions affected of 0.5, 0.75 and 0.9 were averaged for each experiment, and this value was used to calculate the mean between experiments [36].

#### 4.2.2. Evaluation of Intracellular Nucleotide Concentrations by Reversed-Phase High-Performance Liquid Chromatography

MESO-II, STO, H28, H2052 and HMEC-1 cells were seeded in 24-well plates at 50,000 cells per well in 0.5 mL of culture medium (37 °C, 5% CO_2_). After reaching at least 80% confluency, cells were incubated with the inhibitors at concentrations corresponding to the IC_50_ values for each cell line. HMEC-1 cells were treated at concentrations similar to those observed in MM cells. The control cells were treated only with DMSO. After 24 h of treatment, the medium was collected, each well was washed twice with Hanks’ Balanced Salt Solution (HBSS; Gibco, Thermo Fisher Scientific, Waltham, MA, USA), and 300 µL of 0.4 M ice-cold perchloric acid was added. The plate was frozen at −80 °C for at least 24 h, followed by two cycles of defrosting and freezing. The supernatant was collected and centrifuged (10 min, 4 °C, 14,000 rpm), then neutralized to pH 6.5 with 3 M K_3_PO_4_. Intracellular nucleotide concentrations were measured by reversed-phase high-performance liquid chromatography (RP-HPLC) as described previously [37]. The protein levels were determined by the Bradford method in the residue from the plate.

#### 4.2.3. Analysis of Glycolysis and Cell Mito Stress Tests Using the Seahorse Analyzer

To perform the Glycolysis and Cell Mito Stress tests we used the Agilent Seahorse XFp Analyzer (Agilent, Santa Clara, CA, USA) as described previously [38]. For these tests, we selected the most and least resistant mesothelioma cells (H2052 and MESO-II cells, respectively) according to their IC_50_ values. These cells were seeded at a density of 5000 cells per well in an Agilent Seahorse microplate and cultured until reaching about 80% confluency (80 µL culture medium/well, 37 °C, 5% CO_2_). Cells were incubated with DMSO, PGL14, NHI-2 and the combination of those inhibitors at their IC_50_ concentrations for 24 h in normoxia (*n* = 6). Before the Glycolysis Stress Test, the medium was changed to XFp assay medium supplemented with 2 mM glutamine (Agilent, Santa Clara, CA, USA). In order to perform the analysis, 10 mM glucose, 1.0 µM oligomycin and 50 µM 2-DG were added sequentially. For the Cell Mito Stress Test, the medium was changed before analysis to the XFp assay medium with 10 mM glucose, 2 mM glutamine and 1 mM pyruvate (Agilent, Santa Calra, CA, USA). Subsequently, 1.5 µM oligomycin, 1 µM (MESO-II) or 2 µM (H2052) carbonyl cyanide 4-(trifluoromethoxy) phenylhydrazone (FCCP) and 0.5 µM rotenone with antimycin were added sequentially. The XFp sensor cartridge, with the compounds used in the test, was incubated at 37 °C in the non-CO_2_ incubator for 15 min before the assay and the microplate was incubated with the cells for 45 min (after the medium was exchanged). Glycolysis and mitochondrial respiration parameters were calculated according to the manufacturer’s instructions.

#### 4.2.4. Statistics

The experiments were performed (at least) in triplicate and repeated three times, and results are expressed as mean values ± standard error of the mean (SEM). Statistical analysis was performed using one-way or two-way ANOVA with Holm–Sidak post hoc tests using Graph Pad Prism version 9 (Intuitive Software for Science, San Diego, CA, USA).

## 5. Conclusions

GLUT-1 and LDH-A inhibitors were cytotoxic against MM cells and their combination resulted in a synergistic effect but had a minor influence on mitochondrial respiration. Importantly, only slight changes occur in non-cancer cells. To our knowledge, this is the first study that evaluated the intracellular nucleotide concentration in human MM cells, showing major changes after exposure to our LDH-A and GLUT-1 inhibitors. These results, combined with the additional data obtained through the Seahorse analyses, provide a better understanding of the mechanism of action of these inhibitors and a better knowledge of energy metabolism in such an aggressive and resistant cancer. Lastly, our findings support further investigations to develop these compounds as a potential new therapeutic strategy for the treatment of MM.

## Figures and Tables

**Figure 1 ijms-24-07771-f001:**
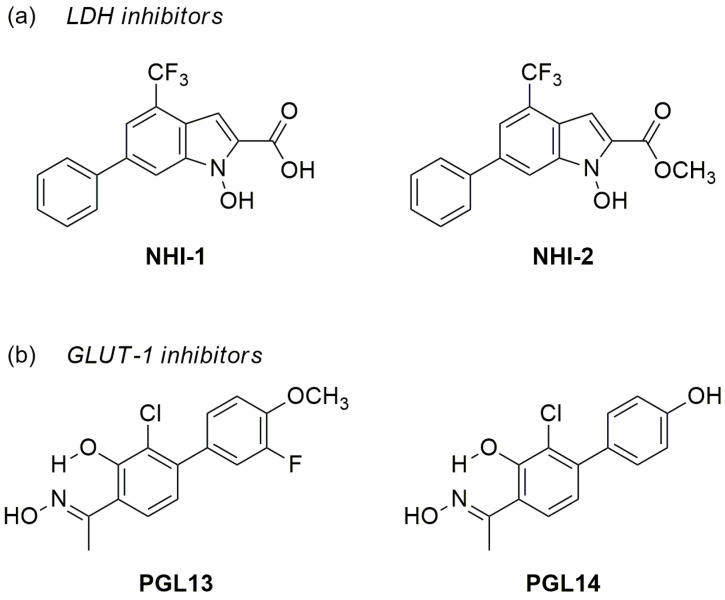
Chemical structures of lactate dehydrogenase A (LDH-A); (**a**) and glucose transporter type 1 (GLUT-1); (**b**) inhibitors.

**Figure 2 ijms-24-07771-f002:**
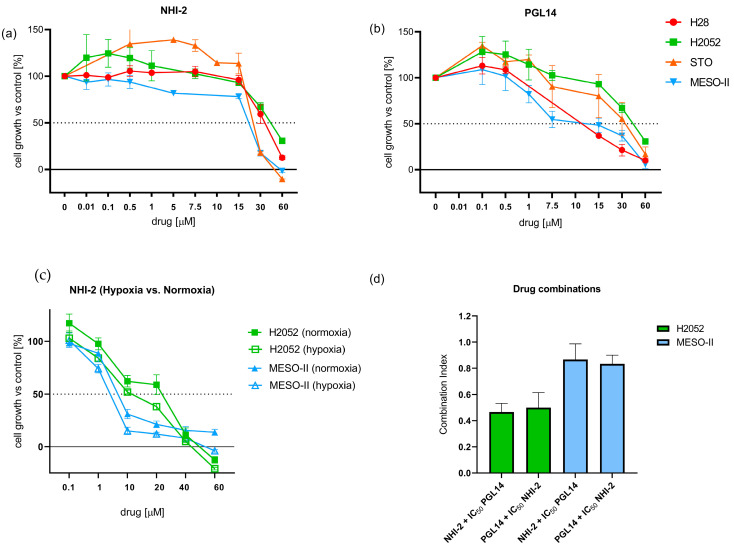
Inhibition of cell growth by LDH-A (NHI-2) and GLUT-1 (PGL14) inhibitors and their combinations. Growth inhibition curves with NHI-2 (**a**) and PGL14 (**b**) in malignant mesothelioma cell lines after 72 h treatment in normoxia and in hypoxia, (**c**) and evaluation of pharmacological interaction using combination index values (**d**). Results are presented as means ± SEM; *n* = 3. LDH-A: lactate dehydrogenase A; GLUT-1: glucose transporter type 1; IC_50_: 50% cell growth inhibitory effect.

**Figure 3 ijms-24-07771-f003:**
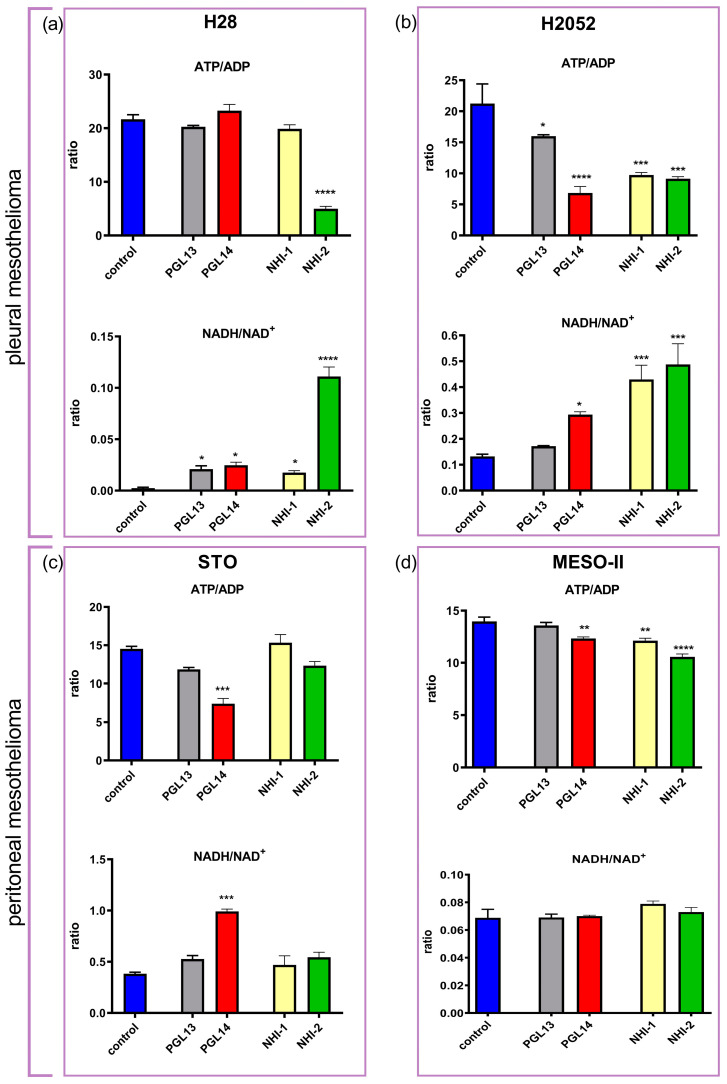
Evaluation of energy charge (ATP/ADP and NADH/NAD^+^ ratios) in pleural (**a**,**b**) and peritoneal (**c**,**d**) mesothelioma cell lines after 24 h exposure to GLUT-1 (PGL13, PGL14) and LDH-A (NHI-1, NHI-2) inhibitors at their IC_50_ concentrations. Ratios were calculated per single experiment and subsequently used to calculate the means. Results are presented as means ± SEM; *n* = 4, one-way ANOVA. * *p* < 0.05; ** *p* < 0.01, *** *p* < 0.005, **** *p* < 0.0001. LDH-A: lactate dehydrogenase A; GLUT-1: glucose transporter type 1; ATP: adenosine triphosphate; ADP: adenosine diphosphate; NADH/NAD^+^: reduced/oxidized nicotinamide adenine dinucleotide; IC_50_: 50% cell growth inhibition concentration.

**Figure 4 ijms-24-07771-f004:**
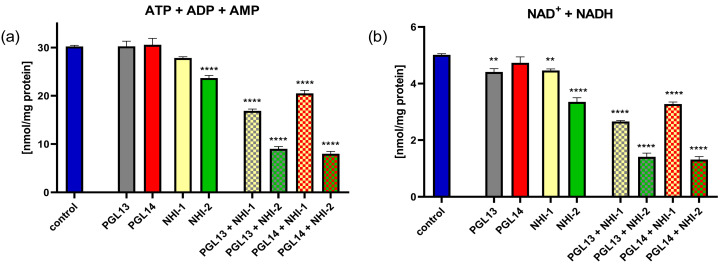
Intracellular purine nucleotide pool (**a**) and nicotinamide adenine dinucleotide pool (**b**) in MESO-II cells after 24 h incubation with GLUT-1 and LDH-A inhibitors alone (IC_50_) and in combination (using IC_50_ of each inhibitor). Results are presented as means ± SEM in nmol/mg protein; *n* = 4, one-way ANOVA. ** *p* < 0.01, **** *p* < 0.0001. LDH-A: lactate dehydrogenase A; GLUT-1: glucose transporter type 1; ATP: adenosine triphosphate; ADP: adenosine diphosphate; AMP: adenosine monophosphate NADH/NAD^+^: reduced/oxidized nicotinamide adenine dinucleotide; IC_50_: 50% cell growth inhibition concentration.

**Figure 5 ijms-24-07771-f005:**
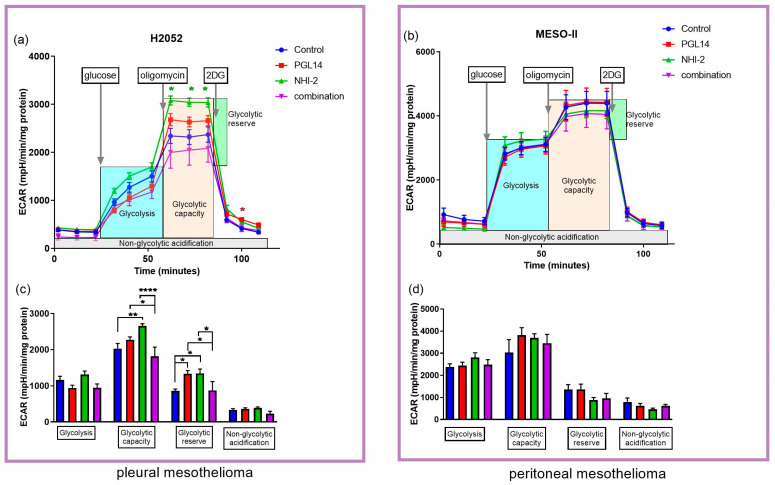
Extracellular acidification rate (ECAR) and calculated glycolysis parameters based on the mean of each separate experiment with H2052 (**a**,**c**) and MESO-II (**b**,**d**) cells after 24 h incubation with inhibitors. 2 DG: 2-deoxy-glucose. Results are presented as mean ± SEM; *n* = 6, two-way ANOVA. * *p* < 0.05; ** *p* < 0.01; **** *p* < 0.0001.

**Figure 6 ijms-24-07771-f006:**
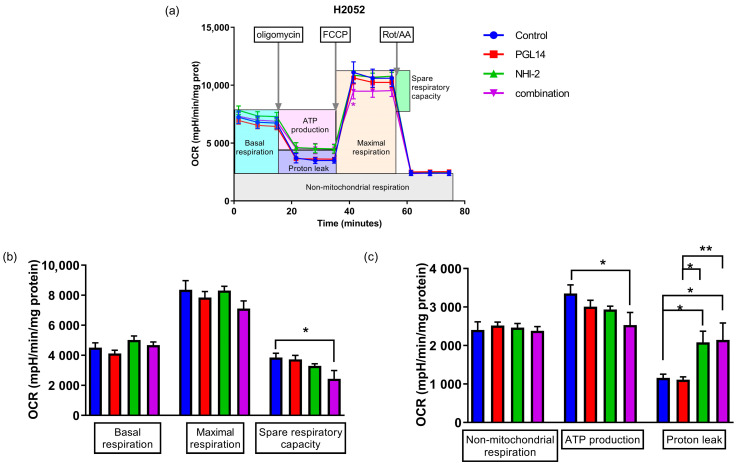
Oxygen consumption rate (OCR); (**a**) and calculated mitochondrial parameters (**b**,**c**) based on the mean of each separate experiment with H2052 cells after 24 h incubation with inhibitors. FCCP: Carbonyl cyanide-4 (trifluoromethoxy) phenylhydrazone; Rot/AA: Rotenone and Antimycin A. Results are presented as mean ± SEM; *n* = 6, two-way ANOVA. * *p* < 0.05; ** *p* < 0.01.

**Figure 7 ijms-24-07771-f007:**
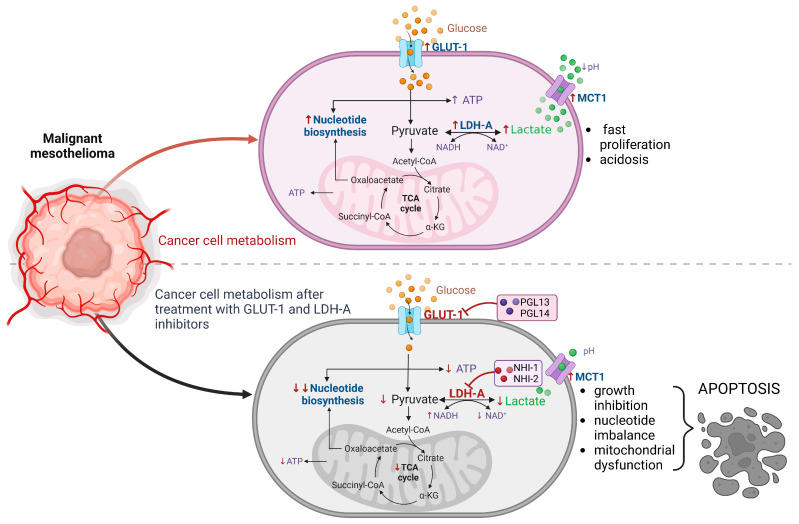
Comparison of the basic metabolism of cancer cells with changes in their metabolism occurring after treatment with GLUT-1 and LDH-A inhibitors. Cancer cell metabolism is characterized by high demand for energy and, as a result, in cancer cells, there is an increased expression of type 1 glucose transporter (GLUT-1), the main energy fuel, as well as of lactate dehydrogenase A (LDH-A), the enzyme that catalyzes the last stage of glycolysis, leading to increased production of lactic acid (transported by monocarboxylate transporter 1: MCT1) and acidification of cells. This allows the cells to have increased production of nucleotides, which in turn helps the cells to proliferate rapidly. Our results show that treating the cells with GLUT-1 (PGL13, PGL14) and LDH-A (NHI-1, NHI-2) inhibitors leads to several changes in their metabolism such as growth inhibition, nucleotide imbalance and mitochondrial dysfunction, that might lead to apoptosis. ↑: high; ↓: low; ATP: adenosine triphosphate; NADH/NAD^+^: reduced/oxidized nicotinamide adenine dinucleotide; TCA cycle: tricarboxylic acid cycle; α -KG: α-ketoglutarate (Created with BioRender.com).

## Data Availability

Not applicable.

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
