# Peer review of "Metabolic Effects of New Glucose Transporter (GLUT-1) and Lactate Dehydrogenase-A (LDH-A) Inhibitors against Chemoresistant Malignant Mesothelioma"

_ijms, 2023, doi:10.3390/ijms24097771_

Round 1

Reviewer 1 Report

This is an interesting and original study, carried out and written to a high scientific standard. The results are important in the context of the urgent unmet medical need in mesothelioma treatment, and the potential for synergistic effects of new molecules modulating tumour metabolism. Since this is a molecular sciences journal, I recommend that the authors give more information on the discovery of the GLUT-1 and LDH-A targeted drug candidates within the main paper (beyond citing the relevant references). This could include providing the chemical structures of the four drug candidates on which this paper was based.

Author Response

This is an interesting and original study, carried out and written to a high scientific standard. The results are important in the context of the urgent unmet medical need in mesothelioma treatment, and the potential for synergistic effects of new molecules modulating tumour metabolism.

Answer: We appreciate the positive summary of this Reviewer

R1.1. Since this is a molecular sciences journal, I recommend that the authors give more information on the discovery of the GLUT-1 and LDH-A targeted drug candidates within the main paper (beyond citing the relevant references). This could include providing the chemical structures of the four drug candidates on which this paper was based.

Answer: We appreciate this important comment which brought our attention to describe drugs more precisely. We added more precise information about the used drugs and added a figure showing their chemical structure in the Introduction section of the revised paper:

The two LDH-A inhibitors used in this study (NHI-1 and NHI-2, Figure 1a) belong to the chemical class of N-hydroxyindole-2-carboxylates, whereas the two GLUT-1 inhibitors (PGL13 and PGL14, Figure 1b) are salicylketoxime derivatives. (lines 93 – 96)

Figure 1. Chemical structures of lactate dehydrogenase A (LDH-A; a) and glucose transporter type 1 (GLUT-1; b) inhibitors.

Reviewer 2 Report

The manuscript entitled "Metabolic effects of new glucose transporter (GLUT-1) and lactate dehydrogenase-A (LDH-A) inhibitors against chemoresistant malignant mesothelioma " in which the authors targeted the the energy metabolism in cancer cells by performing their studies on the cytotoxicity of new inhibitors of glucose transporter type-I (GLUT-1) and lactate dehydrogenase-A (LDH-A) in relation to ATP/NAD+ metabolism, glycolysis and mitochondrial respiration.

The work is understandable and the topic is important and appropriate for publication in International Journal of Molecular Sciences. The work is original, and it contains new results. The abstract describes the essential information in the work. The results, supported by figures, are informative. However, the paper suffers from few shortcomings.

Minor concern:

·         Please define the abbreviations in full name in each figure legend.

·         Please add type of ANOVA used (one-way or two-way ANOVA) in each figure legend.

·         The authors wrote “the human mesothelioma peritoneum primary were derived from samples of patients who underwent surgery, and provided by Dr. Zaffaroni ….etc. Did their experiments need an informant consent from those patients?  and did the authors have an approval from Research ethics committee in their center?

·         I am query, why is there difference in sample number in figure legends? It is sometimes 3, 4 or 6. Please clarify the reason.

Author Response

The manuscript entitled "Metabolic effects of new glucose transporter (GLUT-1) and lactate dehydrogenase-A (LDH-A) inhibitors against chemoresistant malignant mesothelioma " in which the authors targeted the energy metabolism in cancer cells by performing their studies on the cytotoxicity of new inhibitors of glucose transporter type-I (GLUT-1) and lactate dehydrogenase-A (LDH-A) in relation to ATP/NAD+ metabolism, glycolysis and mitochondrial respiration. The work is understandable and the topic is important and appropriate for publication in International Journal of Molecular Sciences. The work is original, and it contains new results. The abstract describes the essential information in the work. The results, supported by figures, are informative. However, the paper suffers from few shortcomings.

Answer: We very much appreciate the positive comments of this Reviewer.

R2.1 Please define the abbreviations in full name in each figure legend.

R2.2 Please add type of ANOVA used (one-way or two-way ANOVA) in each figure legend.

Answer: We thank the Reviewer for pointing out that we omitted the full names of the used abbreviations in the legend of the figures and information of ANOVA type.

We added that information:

Figure 2. (…) LDH-A: lactate dehydrogenase A; GLUT-1: glucose transporter type 1; IC50: 50% cell growth inhibitory effect. (line 131)

Figure 3. (…) one-way ANOVA * p<0.05; ** p<0.01, *** p<0.005, **** p<0.0001. LDH-A: lactate dehydrogenase A; GLUT-1: glucose transporter type 1; ATP: adenosine triphosphate; ADP: adenosine diphosphate; NADH/NAD+: reduce/oxidized nicotinamide adenine dinucleotide; IC50: 50% cell growth inhibition concentration. (lines 153-155)

Figure 4. (…) LDH-A: lactate dehydrogenase A; GLUT-1: glucose transporter type 1; ATP: adenosine triphosphate; ADP: adenosine diphosphate; AMP: adenosine monophosphate NADH/NAD+: reduce/oxidized nicotinamide adenine dinucleotide; IC50: 50% cell growth inhibition concentration. (lines 172-174)

Figure 5. (…) two-way ANOVA * p<0.05;** p<0.01; *** p<0.005; **** p<0.0001. (line 196)

Figure 6. (…) two-way ANOVA * p<0.05; ** p<0.01. (line 209)

Figure 7. (…)ATP: adenosine triphosphate; NADH/NAD+: reduce/oxidized nicotinamide adenine dinucleotide; TCA cycle: tricarboxylic acid cycle; α -KG: α-ketoglutarate. (lines 273-274)

Supplementary Table 1. (…) one-way ANOVA * p<0.05; ** p<0.01, *** p<0.005, **** p<0.0001. LDH-A: lactate dehydrogenase A; GLUT-1: glucose transporter type 1; MM: malignant mesothelioma; IC50: 50% cell growth inhibition concentration; ATP: adenosine triphosphate; ADP: adenosine diphosphate; AMP: adenosine monophosphate NADH/NAD+: reduce/oxidized nicotinamide adenine dinucleotide. (line 3)

Supplementary Figure 2. (…) one-way ANOVA * p<0.05; ** p<0.01, *** p<0.005, **** p<0.0001. GLUT-1: glucose transporter type 1; ATP: adenosine triphosphate; ADP: adenosine diphosphate; AMP: adenosine monophosphate NADH/NAD+: reduce/oxidized nicotinamide adenine dinucleotide. (lines 10-13)

Supplementary Figure 3. (…) one-way ANOVA * p<0.05; ** p<0.01, *** p<0.005, **** p<0.0001. LDH-A: lactate dehydrogenase A; ATP: adenosine triphosphate; ADP: adenosine diphosphate; AMP: adenosine monophosphate NADH/NAD+: reduce/oxidized nicotinamide adenine dinucleotide. (lines 17-19)

R2.3 The authors wrote “the human mesothelioma peritoneum primary were derived from samples of patients who underwent surgery, and provided by Dr. Zaffaroni ….etc. Did their experiments need an informant consent from those patients?  and did the authors have an approval from Research ethics committee in their center?

Answer: We thank the Reviewer for this valuable comment. We added the ethical information in the Materials and Methods section:

(…) the use of the patient samples to generate cell lines was approved by the Institutional Review Board of Fondazione IRCCS Istituto Nazionale dei Tumori (INT). Written informed consent was obtained from all patients to donate the leftover tissue to INT after diagnostic and clinical procedures. (lines 287-290)

R2.4 I am query, why is there difference in sample number in figure legends? It is sometimes 3, 4 or 6. Please clarify the reason.

Answer: We thank the Reviewer for the observation. Each experiment was carefully designed, considering an appropriate number of replications to achieve a reproducible and reliable result. Therefore, these differences are intentional and result from a variety of tests.